# Association between Ambient Air Pollution and Hospital Length of Stay among Children with Asthma in South Texas

**DOI:** 10.3390/ijerph17113812

**Published:** 2020-05-27

**Authors:** Juha Baek, Bita A. Kash, Xiaohui Xu, Mark Benden, Jon Roberts, Genny Carrillo

**Affiliations:** 1Department of Environmental and Occupational Health, School of Public Health, Texas A&M University, College Station, TX 77843, USA; jbaek@tamu.edu (J.B.); mthowdy@tamu.edu (M.B.); 2Center for Outcomes Research, Houston Methodist Research Institute, Houston, TX 77030, USA; bakash@tamu.edu; 3Center for Health and Nature, Houston Methodist Research Institute, Houston, TX 77030, USA; 4Department of Health Policy and Management, School of Public Health, Texas A&M University, College Station, TX 77843, USA; 5Department of Epidemiology and Biostatistics, School of Public Health, Texas A&M University, College Station, TX 77843, USA; xiaohui.xu@tamu.edu; 6Department of Pediatric Pulmonology, Driscoll Children’s Hospital, Corpus Christi, TX 78411, USA; Jon.Roberts@dchstx.org

**Keywords:** ambient air pollution, hospital length of stay, PM_2.5_, ozone, pediatric asthma, South Texas

## Abstract

Although hospital length of stay (LOS) has been identified as a proxy measure of healthcare expenditures in the United States, there are limited studies investigating the potentially important association between outdoor air pollution and LOS for pediatric asthma. This study aims to examine the effect of ambient air pollution on LOS among children with asthma in South Texas. It included retrospective data on 711 children aged 5–18 years old admitted for asthma to a pediatric tertiary care hospital in South Texas between 2010 and 2014. Air pollution data including particulate matter (PM_2.5_) and ozone were collected from the U.S. Centers for Disease Control and Prevention. The multivariate binomial logistic regression analyses were performed to determine the association between each air pollutant and LOS, controlling for confounders. The regression models showed the increased ozone level was significantly associated with prolonged LOS in the single- and two-pollutant models (*p* < 0.05). Furthermore, in the age-stratified models, PM_2.5_ was positively associated with LOS among children aged 5–11 years old (*p* < 0.05). In conclusion, this study revealed a concerning association between ambient air pollution and LOS for pediatric asthma in South Texas.

## 1. Introduction

Hospital length of stay (LOS) is a significant determinant of overall healthcare expenses, and often viewed as a proxy for cost of care. As such, increased LOS causes a substantial economic burden on patients/families and health insurance, including the government [1]. Despite improvements in medical care and medication, the median hospital LOS for pediatric asthma has not changed significantly for the past decades [2]. Additionally, hospital LOS is frequently regarded as an important measurement of healthcare efficiency and resource utilization and greatly affects healthcare planning, hospital capacity, and policy [3,4]. 

Identifying factors that affect hospital LOS is crucial in order to improve health outcomes and effective use of healthcare resources as well as reduce healthcare costs [4]. Several studies have examined the determinants of hospital LOS among patients with asthma. Two studies that investigated all age groups revealed that age, gender, race/ethnicity, admission day, and season were significant factors that influence hospital LOS [5,6]. The other studies on children with asthma found that gender, obesity status, complex chronic conditions, and season were significantly associated with prolonged hospital LOS [7,8,9]. 

In recent years, there is a growing body of literature that examines the relationship between air pollution and asthma hospitalizations. A systematic review found 87 time-series or case-crossover studies that evaluated the association between exposures to outdoor air pollutants and asthma exacerbation outcomes, including hospital admissions and emergency room visits. The study’s meta-analysis found that six major air pollutants, namely ozone, NO_2_, SO_2_, PM_2.5_, PM_10_, and CO, were significantly associated with an increased risk of asthma-related hospitalization [10]. 

Although air pollution affects the health of all age groups, children are more vulnerable to respiratory effects of air pollutants than adults since children are still in developmental and physiologic stages [11,12,13,14,15]. In addition, children are highly exposed to air pollutants because they tend to spend more time outside playing and engaging in physical activity [16,17]. Accordingly, several international studies have explored the effect of air pollution on pediatric asthma hospitalization. A study focusing on children in Vietnam found that short-term exposure to air pollutants including NO_2_ were statistically significant with increased daily counts of hospitalizations due to asthma and bronchitis [18]. A recent study conducted in Taiwan also reported that air pollutants, including PM_10_, PM_2.5_, and SO_2_, were positively associated with hospitalizations among children with asthma [19]. 

Despite the general utility of LOS as a healthcare outcome, only a few studies have explored the relationship between ambient air pollution and hospital LOS among patients with asthma or other respiratory diseases. A study using the 1999–2007 United States (U.S.) National Inpatient Sample data found that PM_2.5_ exposure was significantly associated with the total costs and charges in pediatric asthma hospitalization, but the study did not find any significant relationships between outdoor air pollutants and hospital LOS [20]. On the other hand, significant results were observed in two studies conducted in Asian countries. One study for adults with asthma in China found that ambient air pollutants, including PM_2.5_ and NO_2_, were significantly associated with increased LOS in a stratifying subgroup analysis for sex, age, and season [4]. The other study, conducted in Hanoi, Vietnam, showed a significantly positive association between exposure to ozone and hospital LOS among children with acute lower-respiratory infection (ALRI) [21]. 

Despite the serious public health burden of pediatric asthma hospitalization in the U.S., there are limited studies that investigated the relationship between outdoor air pollution and hospital LOS among children with asthma in the U.S. setting, especially in low-income communities. Therefore, the purpose of this study is to examine the effects of ambient air pollution prior to hospitalization on hospital LOS among pediatric patients with asthma in South Texas, a region with pronounced low-income communities, health disparities, and known environmental health issues [22]. 

## 2. Materials and Methods

### 2.1. Study Setting and Data Sources

Hospitalization records were obtained from the Driscoll Children’s Hospital electronic database to analyze cases of children aged 18 years old or younger with a primary diagnosis of asthma admitted between 1 January 2010 and 31 December 2014. This hospital, located in Corpus Christi, Texas, is a tertiary medical center with 189 beds for pediatric patients and more than 30 medical and surgical specialties that provide healthcare to children living in South Texas [23]. All diagnoses were coded using the International Classification of Diseases, 9th Revision (ICD 9) at discharge (codes 493.0–493.92). Records included information such as age, gender, ethnicity, type of insurance, admission date, discharge date, family history of asthma or respiratory disease, experience of asthma education, use of medication, and census tract information of each child’s residence (i.e., a geographical unit of analysis larger than city block but smaller than city which constitutes the U.S. Census Bureau’s construct for neighborhood level) [24,25]. 

The average daily air pollution concentration data, including particulate matter (PM_2.5_) and ozone, between 1 January 2010 and 31 December 2014, were collected from the U.S. Centers for Disease Control and Prevention (CDC) National Environmental Public Health Tracking Network [26]. These data include the estimates of the mean modeled predictions of PM_2.5_ and ozone concentrations in the census tract level developed by the Downscaler model of the U.S. Environmental Protection Agency (EPA) [27]. These air pollution data are valuable since they estimate the predictions of the two air pollutants in all census tracts in the nation, excluding Hawaii and Alaska. The data also include the areas that do not have air monitoring sites, as well as the daily average modeled concentration levels between 2001 and 2014. This is important given that most monitoring sites do not take samples for PM_2.5_ and ozone on a daily basis [27]. 

The meteorological data (daily mean temperatures) were obtained from the Texas Commission on Environmental Quality (TCEQ) to control for the impact of weather on the LOS for patients with asthma. The temperature data were collected based on the information measured in the nearest air monitoring station from each patient’s residence by using the geographic information system (GIS) program (ArcMap 10.4, ESRI, Redlands, CA, USA). This study protocol was reviewed and approved by the Institutional Review Boards of the Texas A&M University and Driscoll Children’s Hospital. 

### 2.2. Measurement

Hospital length of stay (LOS), the outcome variable of this study, was defined as the total number of nights spent in the hospital from the admission date to the discharge date. In the analysis, this variable was dichotomized as (1) two nights or fewer and (2) more than two nights based on the median LOS for the study population. Ambient air pollution data for PM_2.5_ and ozone concentration levels were the primary independent variables. The daily mean PM_2.5_ and ozone levels from admission day to seven days before admission for each patient were collected individually based on their admission dates and residential census tract information. The data for PM_2.5_ refers to the mean estimated 24 h average concentration in µg/m^3^ and the data for ozone indicates the mean estimated 8 h average concentration in parts per billion (ppb) within three meters of the surface of the earth [27]. 

The effects of air pollutants were measured with different lag days from Lag0 (admission day) to Lag0–7. For example, Lag0–7 represents the eight-day moving average of air pollutant concentrations between admission day and the seventh day before admission. The moving averages were used to evaluate the cumulative effects of air pollutants on hospital LOS [4]. The temperature variable was also measured as the moving averages of daily mean temperatures for the same periods as for the air pollution metric for each patient. 

Furthermore, age, gender, ethnicity, type of insurance, season, and admission day have been identified as significant factors that affect LOS for patients with asthma in previous literature [5,6,7,28]. Other factors associated with asthma control or exacerbation, including family history of asthma, use of medication, asthma education, and outdoor temperature, also may affect the relationship between outdoor air pollution and LOS [29,30,31,32]. Accordingly, potential confounders were included in the regression models as follows: age when admitted to the hospital (5–11 years old or 12–18 years old), gender (male or female), ethnicity (Hispanic or non-Hispanic), type of insurance (public via U.S. Medicaid, private, or self-pay), family history of asthma or other respiratory disease (yes or no), use of medication (yes or no), experience of asthma education (yes or no), season (warm defined as May to October or cold defined as November to April), admission day (weekday as Monday to Thursday or weekend as Friday to Sunday), and outdoor temperature (moving averages as noted above, in Celsius).

### 2.3. Statistical Analysis

Descriptive statistics of the study population were calculated to estimate the mean, standard deviation (SD), and the minimum and maximum for continuous variables or percentages for categorical variables. Pearson correlation was used to assess whether PM_2.5_ and ozone pollutants are highly correlated. In addition, multivariate binomial logistic regression analyses were performed to determine the association between exposure to each air pollutant and hospital LOS on the day of admission (Lag0) and individual cumulative days prior to the admission day (Lag0–1~Lag0–7). The regression models controlled for the other covariates noted above. 

In addition to the single-pollutant regression models, the two-pollutant models including both PM_2.5_ and ozone levels were used to adjust for the mutual effect of air pollutants. The odds ratios (ORs) and 95% confidence intervals (CIs) were estimated for associations between short-term exposure to air pollution on the days prior to admission and hospital LOS. Stratified analyses by age, gender, and season were performed in order to evaluate the effect of confounding factors on the association between air pollutants and LOS for pediatric asthma. All statistical analyses were conducted by using the Stata 14 version (StataCorp LLC, College Station, TX, USA). A p-value less than 0.05 was considered statistically significant. 

## 3. Results

Table 1 shows descriptive statistics of the study population (*N* = 711). The average age of the total participants was nine years old, ranging from five to 18 years old, and three quarters (75.4%) were between five to 11 years old. The low LOS group (≤two nights) had a slightly greater proportion of five to 11 year-old children, compared to the high LOS group (>two nights). The study population consisted of more males (59.1%) than females (40.9%), and the proportion was similar in both LOS groups. About 74% of the children were Hispanic, having a higher rate in the high LOS group (77.4% vs. 73.3%). Over two-thirds of the study population had public insurance (68.7%), and most of the participants (92%) used some form of medication for their asthma care. About half had a family history of asthma or respiratory diseases (49.5%), and this showed to be particularly higher among the high LOS group (56.2%) than the low LOS group (47.6%). Almost all of the study population had received asthma education (95.2%) in the past, and over 60% of them were admitted to the hospital on a weekday (60.6%) and in the cold season (61.6%). None of the characteristics between the two groups were significantly different. 

Summary statistics of daily average air pollutant concentration levels (PM_2.5_ and ozone) and temperatures for the study population from Lag0 to Lag0–7 are shown in Table 2. The average values for PM_2.5_, ozone, and temperatures were similar among the cumulative lag days. However, the minimum and maximum values for each one were different, indicating that the closer the cumulative days are to admission day (such as Lag0, Lag0–1, and Lag0–2), the larger the variations are for the values. For example, the variation for average PM_2.5_ values in Lag0 was about 25 (2.48–27.28), which was higher than the variation (about 12) on Lag0–7 (4.38–16.01). The same patterns were evident among the average values for ozone and temperatures. The Pearson correlation tests between PM_2.5_ and ozone from Lag0 to Lag0–7 revealed small correlations ranging from 0.092 to 0.123 (See Appendix A
Table A1). 

Table 3 and Figure 1 illustrate the results of the multivariate binomial logistic regression analysis to examine the associations between ambient air pollution and hospital LOS on the admission day (Lag0) and cumulative several days before the admission day (Lag0–1 to Lag0–7) in the single- and two-pollutant models, adjusting for several confounders. In the single-pollutant models, the increased ozone concentration level was significantly associated with prolonged hospital LOS from Lag0–1 to Lag0–3 (*p* < 0.05). Positive relationships were also consistently found in Lag0–4, Lag0–5, and Lag0–6, but they were not statistically significant. Moreover, in the two-pollutant model, the ozone concentration level showed a significant positive association with LOS on Lag0–2 (*p* = 0.048). However, the PM_2.5_ concentration level did not have any significant association with LOS, although all of the adjusted ORs showed positive relationships between PM_2.5_ concentration level and hospital LOS. 

Table 4 presents the results of the regression analysis to examine the associations between air pollution and hospital LOS stratified by age. In the group of children aged 5–11 years old, the elevated PM_2.5_ concentration level was significantly associated with longer LOS on Lag0–1 in the single-pollutant (*p* = 0.022) and two-pollutant (*p* = 0.035) models. However, the ozone concentration level did not show significant associations with hospital LOS. Additionally, in the group of children aged 12–18 years old, none of the associations were found to be statistically significant in either the single- or two-pollutant models. 

Table 5 demonstrates the results of regression analysis stratified by gender in single- and two-pollutant models, while controlling for several confounders. For females, we did not find significant relationships, although we observed all of the adjusted ORs were positive. Additionally, no significant associations in single- and two-pollutant models were found among males. 

Table 6 describes the results of season-stratified regression models adjusting for several covariates. During the warm season, ozone concentration level was observed to be significantly associated with prolonged LOS, especially from Lag0–2 to Lag0–7 in the single-pollutant models (*p* < 0.05 or *p* < 0.01). Additionally, the positive relationships between ozone and LOS were consistently significant from Lag0–2 to Lag0–5 in the two-pollutant model (*p* < 0.05). Yet, the PM_2.5_ concentration level did not show a significant effect on hospital LOS. Ozone concentration had a significant effect on increased hospital LOS only on Lag0 during the cold season in both single- and two-pollutants models (*p* < 0.05). However, there were no significant associations for PM_2.5_ concentration in the cold season.

## 4. Discussion

In this study, we examined the association between short-term exposure to outdoor air pollution and hospital LOS among children with asthma in South Texas. We found that increased ozone concentration prior to hospital admission was significantly associated with prolonged hospital LOS for pediatric patients with asthma in both single- and two-pollutant models. This result may be explained by the fact that exposure to ozone can adversely affect the respiratory system, including coughing, chest tightness or pain, throat irritation, and airway inflammation, especially exacerbating asthma conditions [33]. The result also supports previous research for pediatric patients with other respiratory diseases, such as pneumonia or ALRI, in different settings. For example, a study showed that high levels of ozone concentration before hospitalization were related to increased LOS among children aged 0–5 years old with ALRI in Hanoi, Vietnam [21]. The other study revealed that ozone had significantly positive effects on LOS in pneumonia hospitalizations among U.S. children less than 18 years of age [34]. 

Further, age-stratified analysis showed that association between ambient air pollution and LOS may differ with age. The current study found that elevated PM_2.5_ concentration was significantly related with prolonged LOS among younger children aged 5–11 years old. However, the same associations were not significant, among older children aged 12–18 years old, although they were positive. The finding of this study is consistent with research reporting that PM_2.5_ was positively associated with LOS among U.S. children with pneumonia [34]. Other studies have also reported on the positive effects of PM_10_ on LOS among Chinese adults with asthma [4] and Vietnamese children aged 2–5 years with ALRI [21], although PM_2.5_ was not significantly associated with LOS. This study confirms that outdoor air pollutants are significantly associated with hospital LOS for at least some pediatric asthma patients in South Texas. 

Conversely, the models stratified by gender demonstrated that ambient air pollutants (PM_2.5_ and ozone) did not show a significant effect on hospital LOS among girls and boys in the single- and two-pollutant models. This study therefore did not observe gender as a modifying factor in the effect of outdoor air pollution on LOS among children with asthma. This finding is contrary to a study that found the positive effect of ambient air pollutants, especially PM_2.5_ concentration, on LOS among females aged 15 years or older in China [4]. Since very few studies examined the impact of gender on the association between outdoor air pollution and hospital LOS, more evidence is still needed to assess the gender differences in the relationship for pediatric patients with respiratory diseases like asthma. 

Season-stratified models revealed a positive correlation of PM_2.5_ and ozone concentration levels on hospital LOS during the warm season and cold season generally, but significant associations were observed only for ozone. Our findings are consistent with a previous study indicating significantly positive associations of particulate matters, including PM_2.5_ and PM_10_, among adults with asthma on LOS during the cold season and warm season [4]. The current study’s results are also in line with those of previous studies that reported on the effects of season on the relationship between outdoor air pollution and hospitalization outcomes for pediatric asthma. For example, most studies found that ozone concentrations had significantly positive effects on asthma hospitalizations or ED visits in the warm season [35,36,37,38]. Additionally, some studies presented that asthma hospital visits peak in the fall season, especially for school-aged children, despite ozone level peaking in the summer season. The explanation has been speculated to be that students transmit respiratory viruses and bacteria to their peers during the school year, causing an asthma attack [39,40,41].

However, in contrast to our findings of no significant relationship for PM_2.5_, some evidence reported significant association between PM_2.5_ and asthma hospital admission (or ED visits) in the warm season [42,43]. Although our results indicate that season might be a modifying factor for the relationship between outdoor air pollution and LOS among people with asthma, additional studies need to be undertaken to confirm the modifying effect of season.

### 4.1. Limitations

There are some limitations to this study. First, we used the estimates of the modeled predictions of ambient air pollution data by the Downscaler approach. Although this approach covered all areas in the census tract level, including regions with no air monitors, it may not accurately reflect the exact air conditions of participants’ residence and personal exposure, particularly in regions with limited monitoring data [44]. Additionally, temperature data collected from a few monitoring sites may not reflect actual temperature due to variations in distances to the nearest monitoring site. As a result, this may lead to measurement error. Second, we included only two air pollutants (PM_2.5_ and ozone) in this study due to limited availability of data. Future study should include other air pollutants, such as PM_10_, NO_2_, and SO_2_, in order to have a better understanding of their relationships with hospital LOS. 

Third, some patient-level factors, such as severity and type of comorbidities (e.g., obesity) [6,9,45], could affect individuals’ hospital LOS, but these factors were unavailable in the hospital database. Future study should consider these factors to control for their effects on the relationship between ambient air pollution and LOS. Fourth, hospital LOS may not always be an accurate outcome measure due to patients’ situations. Particularly in this low-income population, the point at which a patient is ready for discharge sometimes differs from the actual length of stay. Additionally, there are various definitions to operationalize the concept of increased LOS; a gold standard does not exist [46,47]. Yet, previous studies have used hospital LOS that is greater than the mean or median values as prolonged LOS [7,21,48,49]. Lastly, this study was conducted in a single hospital located in South Texas so that generalizability of results to other settings in different regions may be limited. 

### 4.2. Public Health and Policy Implications

The findings of this study have important public health and policy implications. First of all, this study suggests that the importance of outdoor air pollution on asthma control and management should be emphasized in asthma education, particularly for parents and guardians of pediatric patients. Given that asthma education is an effective way to improve knowledge for asthma management and health outcomes of children with asthma [28,50,51], it would be important to supplement and highlight educational contents on outdoor air pollution in the curriculum, such as the different types of ambient air pollutants, their impacts on health outcomes, and practical ways to minimize exposure to harmful outdoor air pollutants. Particularly, the current asthma education contents at Driscoll Children’s Hospital focus on indoor air pollutants and have limited information about outdoor air pollution. Thus, the contents about ambient air pollution need to be included in the asthma education at the hospital. This study also suggests that healthcare professionals, who are in charge of asthma education, actively communicate with children with asthma and their family how to check daily outdoor air pollution levels and explain the importance of limiting outdoor activities on days when air pollutant levels are high. 

In addition, this study will help hospital leaders and pediatricians who serve children with asthma gain a better understanding on how ambient air pollution could be an important indicator for identifying pediatric patients with asthma who have a risk for longer LOS. This may also help inform more effective healthcare resource allocation and utilization, focusing on care for children from regions with high levels of air pollution, in order to decrease hospital LOS. Furthermore, our results may help school leaders understand the effects of outdoor air pollutants on health outcomes of children, especially those with asthma, and consider the outdoor air quality when planning for outdoor activities or school events. Finally, the findings of this study offer important evidence to policymakers who support policies related to ambient air pollution control.

## 5. Conclusions

This study found the adverse effects of outdoor air pollutants, including PM_2.5_ and ozone concentrations, on hospital LOS for pediatric asthma in South Texas, especially in low-income communities. Our results showed that age and season might be modifying factors on the relationship between ambient air pollution and LOS. These findings may help health professionals and policymakers consider the importance of ambient air pollution on health outcomes and hospital LOS among pediatric patients with asthma during education sessions, medical care practice, and policy formulation, emphasizing preventive measures that need to be included in educational programs targeted for children. 

## Figures and Tables

**Figure 1 ijerph-17-03812-f001:**
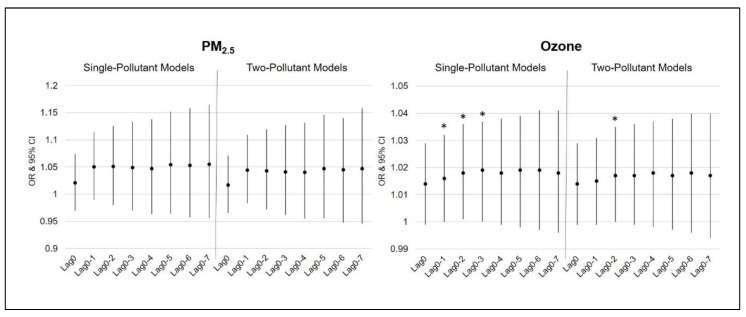
Lag structures of the odds ratio (OR) and 95% confidence interval (CI) of PM_2.5_ and ozone with hospital length of stay in single-pollutant and two-pollutant models. Note: OR—odds ratio; 95% CI—95% confidence interval; * *p* < 0.05.

**Table 1 ijerph-17-03812-t001:** Descriptive statistics of study population (*N* = 711).

Variables	Total ^†^	High (LOS > 2) ^†^	Low (LOS ≤ 2) ^†^	*p*-Value
**Total**	711 (100.0)	160 (100.0)	551 (100.0)	
**Age (continuous)**	9.2 ± 3.5 (5, 18)	9.5 ± 3.6 (5, 18)	9.1 ± 3.5 (5, 18)	0.152
Age				0.451
5–11 years old	536 (75.4)	117 (73.1)	419 (76.0)	
12–18 years old	175 (24.6)	43 (26.9)	132 (24.0)	
**Gender**				0.925
Female	291 (40.9)	66 (41.3)	225 (40.8)	
Male	420 (59.1)	94 (58.7)	326 (59.2)	
**Ethnicity**				0.305
Hispanic	527 (74.2)	123 (77.4)	404 (73.3)	
Non-Hispanic	183 (25.8)	36 (22.6)	147 (26.7)	
**Insurance**				0.733
Public (Medicaid)	488 (68.7)	108 (67.5)	380 (69.0)	
Private	205 (28.8)	49 (30.6)	156 (28.3)	
Self-pay	18 (2.5)	3 (1.9)	15 (2.7)	
**Use of Medication**				0.206
Yes	654 (92.0)	151 (94.4)	503 (91.3)	
No	57 (8.0)	9 (5.6)	48 (8.7)	
**Family History of Asthma or** **Respiratory Diseases**				0.053
Yes	352 (49.5)	90 (56.2)	262 (47.6)	
No	359 (50.5)	70 (43.8)	289 (52.4)	
**Recipient of Asthma Education**				0.265
Yes	677 (95.2)	155 (96.9)	522 (94.7)	
No	34 (4.8)	5 (3.1)	29 (5.3)	
**Admission Day**				0.199
Weekday (Mon –Thu)	431 (60.6)	90 (56.2)	341 (61.9)	
Weekend (Fri–Sun)	280 (39.4)	70 (43.8)	210 (38.1)	
**Admission Season**				0.526
Warm (May–October)	273 (38.4)	58 (36.3)	215 (39.0)	
Cold (November–April)	438 (61.6)	102 (63.7)	336 (61.0)	
**Year**				0.081
2010	166 (23.3)	48 (30.0)	118 (21.4)	
2011	133 (18.7)	34 (21.1)	99 (18.0)	
2012	154 (21.7)	26 (16.3)	128 (23.2)	
2013	125 (17.6)	26 (16.3)	99 (18.0)	
2014	133 (18.7)	26 (16.3)	107 (19.4)	

Note: LOS – Length of Stay; ^†^ Mean ± standard deviation (minimum, maximum) or N (%).

**Table 2 ijerph-17-03812-t002:** Summary statistics of daily average air pollutant concentrations and temperatures from Lag0 to Lag0–7.

Lag Days Pre-Admission	PM_2.5_ (µg/m^3^)				Ozone (ppb)				Temperature (°C)			
	Mean	SD	Min	Max	Mean	SD	Min	Max	Mean	SD	Min	Max
Lag0	8.43	3.41	2.48	27.28	37.43	11.92	10.09	75.23	20.71	6.78	1.06	33.33
Lag0–1	8.45	2.89	2.96	23.01	37.66	11.14	13.74	75.29	20.65	6.64	2.11	32.92
Lag0–2	8.47	2.50	3.57	19.96	37.80	10.42	15.69	73.35	20.65	6.52	1.78	32.41
Lag0–3	8.48	2.24	3.80	17.85	37.84	9.74	16.84	73.58	20.66	6.45	1.02	32.08
Lag0–4	8.47	2.10	3.75	16.46	37.87	9.24	17.39	72.57	20.65	6.37	2.99	32.1
Lag0–5	8.46	1.96	3.88	16.71	37.81	8.85	17.48	71.85	20.63	6.34	5.82	32.15
Lag0–6	8.45	1.85	4.08	16.46	37.72	8.39	17.73	69.02	20.62	6.33	6.33	32.24
Lag0–7	8.47	1.78	4.38	16.01	37.61	8.05	17.93	67.21	20.61	6.33	6.63	32.37

Note: SD—standard deviation; ppb—parts per billion.

**Table 3 ijerph-17-03812-t003:** Results of multivariate binomial logistic regression analysis in single- and two-pollutant models.

Lag Days Pre-Admission.	Single-Pollutant Models				Two-Pollutant Models			
	PM_2.5_		Ozone		PM_2.5_		Ozone	
	OR (95% CI)	*p*-Value	OR (95% CI)	*p*-Value	OR (95% CI)	*p*-Value	OR (95% CI)	*p*-Value
Lag0	1.021 (0.970, 1.074)	0.427	1.014 (0.999, 1.029)	0.059	1.017 (0.965, 1.071)	0.528	1.014 (0.999, 1.029)	0.068
Lag0–1	1.050 (0.989, 1.114)	0.112	1.016 (1.000, 1.032)	0.049 *	1.044 (0.983, 1.109)	0.162	1.015 (0.999, 1.031)	0.069
Lag0–2	1.051 (0.980, 1.126)	0.162	1.018 (1.001, 1.036)	0.033 *	1.043 (0.972, 1.119)	0.247	1.017 (1.000, 1.035)	0.048 *
Lag0–3	1.049 (0.970, 1.134)	0.231	1.019 (1.000, 1.037)	0.048 *	1.041 (0.962, 1.127)	0.317	1.017 (0.999, 1.036)	0.063
Lag0–4	1.047 (0.963, 1.138)	0.279	1.018 (0.999, 1.038)	0.063	1.040 (0.955, 1.131)	0.366	1.018 (0.998, 1.037)	0.079
Lag0–5	1.054 (0.964, 1.152)	0.246	1.019 (0.998, 1.039)	0.076	1.047 (0.956, 1.146)	0.321	1.017 (0.997, 1.038)	0.095
Lag0–6	1.053 (0.957, 1.158)	0.288	1.019 (0.997, 1.041)	0.086	1.045 (0.948, 1.140)	0.376	1.018 (0.996, 1.040)	0.107
Lag0–7	1.055 (0.956, 1.165)	0.287	1.018 (0.996, 1.041)	0.113	1.047 (0.946, 1.158)	0.372	1.017 (0.994, 1.040)	0.141

Note: OR—Odds Ratio, 95% CI—95% Confidence Interval; Adjusted for age, gender, ethnicity, family history of asthma or respiratory diseases, type of insurance, use of medication, experience of asthma education, season, admission day, and temperature. * *p* < 0.05.

**Table 4 ijerph-17-03812-t004:** Results of multivariate binomial logistic regression analysis stratified by age in single- and two-pollutant models.

**Single-Pollutant Models**								
**Lag Days**	**5–11 Years Old**				**12–18 Years Old**			
	**PM_2.5_**		**Ozone**		**PM_2.5_**		**Ozone**	
	**OR (95% CI)**	***p*-Value**	**OR (95% CI)**	***p*-Value**	**OR (95% CI)**	***p*-Value**	**OR (95% CI)**	***p*-Value**
Lag0	1.052 (0.992, 1.116)	0.092	1.015 (0.998, 1.033)	0.091	0.926 (0.821, 1.046)	0.262	1.012 (0.983, 1.042)	0.422
Lag0–1	1.086 (1.012, 1.165)	0.022 *	1.017 (0.998, 1.036)	0.081	0.941 (0.822, 1.076)	0.373	1.014 (0.982, 1.047)	0.403
Lag0–2	1.077 (0.993, 1.167)	0.074	1.019 (0.999, 1.040)	0.063	0.958 (0.821, 1.118)	0.590	1.013 (0.979, 1.048)	0.467
Lag0–3	1.077 (0.984, 1.179)	0.107	1.020 (0.998, 1.043)	0.072	0.930 (0.780, 1.109)	0.422	1.008 (0.972, 1.045)	0.672
Lag0–4	1.079 (0.979, 1.189)	0.124	1.019 (0.996, 1.044)	0.096	0.922 (0.769, 1.104)	0.375	1.008 (0.970, 1.048)	0.670
Lag0–5	1.089 (0.982, 1.209)	0.107	1.018 (0.994, 1.044)	0.148	0.929 (0.771, 1.121)	0.443	1.015 (0.975, 1.056)	0.474
Lag0–6	1.109 (0.975, 1.219)	0.129	1.019 (0.993, 1.046)	0.159	0.922 (0.756, 1.126)	0.426	1.015 (0.973, 1.058)	0.485
Lag0–7	1.095 (0.974, 1.231)	0.127	1.018 (0.991, 1.047)	0.193	0.928 (0.757, 1.138)	0.475	1.014 (0.971, 1.059)	0.525
**Two-Pollutant Models**								
**Lag Days**	**5–11 Years Old**				**12–18 Years Old**			
	**PM_2.5_**		**Ozone**		**PM_2.5_**		**Ozone**	
	**OR (95% CI)**	***p*-Value**	**OR (95% CI)**	***p*-Value**	**OR (95% CI)**	***p*-Value**	**OR (95% CI)**	***p*-Value**
Lag0	1.047 (0.986, 1.112)	0.134	1.014 (0.996, 1.032)	0.131	0.922 (0.816, 1.042)	0.194	1.014 (0.984, 1.045)	0.363
Lag0–1	1.080 (1.005, 1.160)	0.035 *	1.015 (0.996, 1.034)	0.134	0.935 (0.816, 1.071)	0.332	1.016 (0.983, 1.050)	0.354
Lag0–2	1.069 (0.984, 1.160)	0.114	1.017 (0.997, 1.038)	0.096	0.951 (0.814, 1.112)	0.530	1.014 (0.979, 1.050)	0.425
Lag0–3	1.069 (0.975, 1.172)	0.154	1.018 (0.996, 1.041)	0.103	0.928 (0.778, 1.107)	0.408	1.009 (0.972, 1.047)	0.636
Lag0–4	1.071 (0.971, 1.182)	0.171	1.018 (0.995, 1.042)	0.132	0.919 (0.768, 1.102)	0.364	1.010 (0.970, 1.050)	0.637
Lag0–5	1.083 (0.974, 1.204)	0.140	1.016 (0.992, 1.042)	0.197	0.924 (0.765, 1.116)	0.410	1.016 (0.976, 1.059)	0.435
Lag0–6	1.083 (0.967, 1.214)	0.165	1.017 (0.991, 1.044)	0.165	0.912 (0.745, 1.117)	0.374	1.018 (0.975, 1.063)	0.418
Lag0–7	1.090 (0.968, 1.227)	0.153	1.017 (0.989, 1.045)	0.237	0.914 (0.741, 1.127)	0.400	1.018 (0.973, 1.065)	0.433

Note: OR—Odds Ratio, 95% CI—95% Confidence Interval; Adjusted for gender, ethnicity, family history of asthma or respiratory diseases, type of insurance, use of medication, experience of asthma education, season, admission day, and temperature. * *p* < 0.05.

**Table 5 ijerph-17-03812-t005:** Results of multivariate binomial logistic regression analysis stratified by gender in single- and two-pollutant models.

**Single-Pollutant Models**								
**Lag Days**	**Female**				**Male**			
	**PM_2.5_**		**Ozone**		**PM_2.5_**		**Ozone**	
	**OR (95% CI)**	***p*-Value**	**OR (95% CI)**	***p*-Value**	**OR (95% CI)**	***p*-Value**	**OR (95% CI)**	***p*-Value**
Lag0	1.023 (0.944, 1.108)	0.579	1.015 (0.991, 1.038)	0.221	1.021 (0.953, 1.093)	0.557	1.011 (0.992, 1.032)	0.231
Lag0–1	1.076 (0.979, 1.183)	0.130	1.016 (0.991, 1.042)	0.218	1.044 (0.963, 1.132)	0.298	1.015 (0.994, 1.036)	0.168
Lag0–2	1.096 (0.984, 1.221)	0.096	1.020 (0.993, 1.047)	0.141	1.031 (0.937, 1.135)	0.534	1.017 (0.994, 1.041)	0.142
Lag0–3	1.085 (0.961, 1.223)	0.187	1.023 (0.994, 1.052)	0.118	1.034 (0.928, 1.151)	0.546	1.015 (0.990, 1.041)	0.241
Lag0–4	1.093 (0.958, 1.247)	0.184	1.027 (0.996, 1.058)	0.085	1.022 (0.914, 1.143)	0.701	1.012 (0.986, 1.039)	0.367
Lag0–5	1.131 (0.985, 1.299)	0.080	1.030 (0.998, 1.064)	0.067	1.006 (0.890, 1.136)	0.928	1.010 (0.983, 1.038)	0.474
Lag0–6	1.135 (0.983, 1.309)	0.083	1.033 (0.999, 1.069)	0.060	0.995 (0.872, 1.135)	0.940	1.009 (0.981, 1.038)	0.533
Lag0–7	1.106 (0.955, 1.282)	0.180	1.030 (0.994, 1.067)	0.097	1.017 (0.886, 1.167)	0.813	1.010 (0.980, 1.041)	0.517
**Two-Pollutant Models**								
**Lag Days**	**Female**				**Male**			
	**PM_2.5_**		**Ozone**		**PM_2.5_**		**Ozone**	
	**OR (95% CI)**	***p*-Value**	**OR (95% CI)**	***p*-Value**	**OR (95% CI)**	***p*-Value**	**OR (95% CI)**	***p*-Value**
Lag0	1.017 (0.938, 1.103)	0.680	1.039 (0.991, 1.038)	0.224	1.018 (0.949, 1.091)	0.616	1.011 (0.992, 1.032)	0.247
Lag0–1	1.066 (0.968, 1.175)	0.193	1.013 (0.987, 1.039)	0.335	1.042 (0.960, 1.131)	0.326	1.014 (0.993, 1.036)	0.182
Lag0–2	1.083 (0.969, 1.210)	0.160	1.016 (0.989, 1.044)	0.240	1.026 (0.932, 1.131)	0.596	1.017 (0.994, 1.040)	0.153
Lag0–3	1.067 (0.942, 1.208)	0.308	1.019 (0.991, 1.049)	0.187	1.032 (0.926, 1.149)	0.572	1.015 (0.989, 1.040)	0.249
Lag0–4	1.071 (0.933, 1.228)	0.331	1.023 (0.992, 1.055)	0.143	1.021 (0.912, 1.143)	0.716	1.012 (0.986, 1.039)	0.372
Lag0–5	1.107 (0.958, 1.278)	0.168	1.025 (0.992, 1.058)	0.140	1.005 (0.889, 1.136)	0.937	1.010 (0.983, 1.038)	0.475
Lag0–6	1.109 (0.955, 1.287)	0.176	1.027 (0.993, 1.064)	0.124	0.994 (0.870, 1.135)	0.928	1.009 (0.981, 1.039)	0.532
Lag0–7	1.082 (0.927, 1.262)	0.317	1.026 (0.989, 1.063)	0.164	1.016 (0.884, 1.166)	0.828	1.010 (0.980, 1.040)	0.521

Note: OR—Odds Ratio, 95% CI—95% Confidence Interval; Adjusted for age, ethnicity, family history of asthma or respiratory diseases, type of insurance, use of medication, experience of asthma education, season, admission day, and temperature.

**Table 6 ijerph-17-03812-t006:** Results of multivariate binomial logistic regression analysis stratified by season in single- and two-pollutant models.

**Single-Pollutant Models**								
**Lag Days**	**Warm Season**				**Cold Season**			
	**PM_2.5_**		**Ozone**		**PM_2.5_**		**Ozone**	
	**OR (95% CI)**	***p*-Value**	**OR (95% CI)**	***p*-Value**	**OR (95% CI)**	***p*-Value**	**OR (95% CI)**	***p*-Value**
Lag0	1.025 (0.947, 1.109)	0.544	1.014 (0.993, 1.036)	0.198	1.022 (0.954, 1.095)	0.538	1.027 (1.003, 1.053)	0.028 *
Lag0–1	1.046 (0.961, 1.139)	0.298	1.021 (0.999, 1.044)	0.058	1.063 (0.973, 1.161)	0.178	1.022 (0.995, 1.051)	0.115
Lag0–2	1.070 (0.972, 1.178)	0.167	1.027 (1.004, 1.051)	0.020 *	1.035 (0.932, 1.150)	0.520	1.021 (0.990, 1.053)	0.180
Lag0–3	1.094 (0.984, 1.216)	0.097	1.032 (1.007, 1.057)	0.012 *	1.001 (0.887, 1.130)	0.983	1.018 (0.984, 1.053)	0.309
Lag0–4	1.105 (0.987, 1.237)	0.083	1.036 (1.009, 1.063)	0.009 **	0.983 (0.863, 1.120)	0.798	1.015 (0.979, 1.053)	0.421
Lag0–5	1.122 (0.996, 1.265)	0.058	1.036 (1.008, 1.066)	0.012 *	0.977 (0.847, 1.127)	0.750	1.017 (0.978, 1.058)	0.397
Lag0–6	1.121 (0.989, 1.269)	0.073	1.037 (1.006, 1.069)	0.018 *	0.977 (0.835, 1.142)	0.770	1.021 (0.979, 1.064)	0.337
Lag0–7	1.131 (0.995, 1.286)	0.059	1.035 (1.003, 1.068)	0.030 *	0.959 (0.812, 1.132)	0.621	1.025 (0.980, 1.072)	0.283
**Two-Pollutant Models**								
**Lag Days**	**Warm Season**				**Cold Season**			
	**PM_2.5_**		**Ozone**		**PM_2.5_**		**Ozone**	
	**OR (95% CI)**	***p*-Value**	**OR (95% CI)**	***p*-Value**	**OR (95% CI)**	***p*-Value**	**OR (95% CI)**	***p*-Value**
Lag0	1.009 (0.927, 1.099)	0.830	1.013 (0.991, 1.036)	0.251	1.034 (0.964, 1.109)	0.344	1.029 (1.004, 1.055)	0.021 *
Lag0–1	1.023 (0.933, 1.121)	0.633	1.019 (0.996, 1.043)	0.100	1.076 (0.983, 1.178)	0.111	1.026 (0.997, 1.055)	0.075
Lag0–2	1.038 (0.935, 1.152)	0.484	1.024 (1.000, 1.049)	0.047 *	1.043 (0.938, 1.160)	0.432	1.022 (0.991, 1.055)	0.158
Lag0–3	1.058 (0.942, 1.187)	0.343	1.028 (1.002, 1.054)	0.036 *	1.009 (0.893, 1.141)	0.877	1.018 (0.984, 1.054)	0.304
Lag0–4	1.066 (0.942, 1.205)	0.310	1.031 (1.003, 1.059)	0.028 *	0.989 (0.868, 1.129)	0.878	1.015 (0.978, 1.053)	0.436
Lag0–5	1.083 (0.951, 1.232)	0.229	1.030 (1.001, 1.061)	0.043 *	0.985 (0.853, 1.139)	0.843	1.016 (0.977, 1.057)	0.419
Lag0–6	1.081 (0.945, 1.238)	0.253	1.031 (0.999, 1.063)	0.057	0.987 (0.843, 1.157)	0.876	1.020 (0.978, 1.064)	0.354
Lag0–7	1.096 (0.954, 1.257)	0.194	1.028 (0.995, 1.062)	0.096	0.973 (0.822, 1.152)	0.751	1.023 (0.978, 1.071)	0.316

Note: OR—Odds Ratio, 95% CI—95% Confidence Interval; Adjusted for age, gender, ethnicity, family history of asthma or respiratory diseases, type of insurance, use of medication, experience of asthma education, admission day, and temperature. * *p* < 0.05, ** *p* < 0.01.

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
