# Peer review of "Association between Ambient Air Pollution and Hospital Length of Stay among Children with Asthma in South Texas"

_ijerph, 2020, doi:10.3390/ijerph17113812_

Round 1

Reviewer 1 Report

This a potential important manuscript because of the attempt to examine the association between ambient air pollution on the hospital length of stay among children with asthma in low-income communities with known health disparities. The authors presented the results with clarity, and the manuscript topic fits well in the journal. The discussion is well organized with key results, implications including topics needing further study.

minor comments:

1. Abstract: The authors should mention the study setting in the abstract: tertiary care hospital.
2. Table 4: The significant level is 0.05 but, in the text (line 212), the authors used significant levels of 0.05 and 0.1. Please be consistent with the significant level indicated in the method section.
3. Discussion 4.2: Can the authors comment on the current asthma education content at this pediatric hospital? Do the educational materials have information on outdoor air pollution?

Author Response

Comments and Suggestions for Authors

This a potential important manuscript because of the attempt to examine the association between ambient air pollution on the hospital length of stay among children with asthma in low-income communities with known health disparities. The authors presented the results with clarity, and the manuscript topic fits well in the journal. The discussion is well organized with key results, implications including topics needing further study.

minor comments:

  1. Abstract: The authors should mention the study setting in the abstract: tertiary care hospital.

=> In response to your suggestion, we changed “a children’s hospital” to “a pediatric tertiary care hospital” in the abstract.

  1. Table 4: The significant level is 0.05 but, in the text (line 212), the authors used significant levels of 0.05 and 0.1. Please be consistent with the significant level indicated in the method section.

            => Thank you for pointing that out. We revised this part to “However, the ozone concentration level did not show significant associations with hospital LOS.” (Track-change, lines 219-220)

  1. Discussion 4.2: Can the authors comment on the current asthma education content at this pediatric hospital? Do the educational materials have information on outdoor air pollution?

            => We checked the current asthma education content at Driscoll Children’s Hospital and included some information about the contents in the discussion 4.2 section (Track-change, lines 324-327).

“Particularly, the current asthma education contents at Driscoll Children’s Hospital focus on indoor air pollutants and have limited information about outdoor air pollution. Thus, the contents about ambient air pollution need to be included in the asthma education at the hospital.”

Reviewer 2 Report

The authors report adverse effects of outdoor air pollutants, including PM2.5 and ozone concentrations, on hospital LOS for pediatric asthma in South Texas, especially in low-income communities. The age and season might be modifying factors on the relationship between ambient air pollution and LOS. This is an important report for policy makers and health care workers to consider the importance of ambient air pollution on health in young children with asthma with emphasis on PM2.5 and ozone.Minor comments:

1. A graphical abstract with the key finding would increase the visibility of this contribution

2. The abstract and the introduction should shortened.

Author Response

Response to Reviewer 2 Comments

The authors report adverse effects of outdoor air pollutants, including PM2.5 and ozone concentrations, on hospital LOS for pediatric asthma in South Texas, especially in low-income communities. The age and season might be modifying factors on the relationship between ambient air pollution and LOS. This is an important report for policy makers and health care workers to consider the importance of ambient air pollution on health in young children with asthma with emphasis on PM2.5 and ozone. Minor comments:

  1. A graphical abstract with the key finding would increase the visibility of this contribution

            => Thank you for your suggestion. In response to that, we created a figure (Figure 1) showing the results of multivariate binomial logistic regression analysis in single- and two-pollutant models and added it for a graphical abstract in this paper.

  1. The abstract and the introduction should shortened.

            => In response to your suggestion, we shortened the abstract and the introduction.

Reviewer 3 Report

This paper by Baek et al assesses the relationship between PM2.5 and O3 in a series of control tests of children with asthma and hospital length of stay using four years of hospital records in a S. Texas community.

The manuscript was very well prepared, the analysis was sound, the presentation was thorough, and the discussion had a good balance of implications and assessment of the data presented. It was very clear the authors put an exhaustive amount of effort into this paper submission. I do have a few minor suggestions that I think, if incorporated, can improve the manuscript.

 Line 68: I did a quick google scholar search and found a number of other studies examining the relationship between air pollution and LOS. Maybe rephrase this to “only a few studies”

Line 124-127: Is the ozone the maximum daily 8-hr average ozone? I don’t understand the utility of an 8-hr average concentration in this analysis – was it then combined to give a daily average?

Table 1 The Female-Low LOS cell is missing a “)”

Line180: It’s important to indicate that the NAAQs have different averaging times than the data that is presented here. It’s not really an apples-to-apples comparison.

Table2-6: This is nitpicky, but can you include vertical lines between the pollutants? It may make the tables easier to read.

Line 210 (and throughout): Please make sure to include mention of the alpha levels on the statistical tests. The finding here, specifically, is insignificant at 0.05, but is significant at 0.10.

Line 248-249: I don’t see how this conclusion was made from the data presented when the analysis was only for children in the first place.

Line 263: I think “assess” or something to that affect Is more appropriate than “confirm”

Line 265-266: I don’t think this statement represents the data in Table 6 completely. Sure, there aren’t any significant associations for PM2.5 but the OR is generally positive during the warm and cold season for PM2.5 as well. I just think this sentence should be rephrased.

Line 268: comma after PM10

Sentences starting at 269: There is a large body of literature that finds childhood asthma hospital visits actually peaks in the Fall, despite O3 peaking in the hottest parts of summer. This has been attributed to kids being in school with other children who may transfer viruses and bacteria that can trigger an asthma attack. I think this is important context that should be included in the discussion here.

Line 314-317: While I agree with the statement, I don’t see this as an appropriate extension from the data presented here. Short term exposures (which would be most outdoor activities for schoolkids), proxied by the lag0 data, have some of the lowest OR correlations

Line 321-322: Where did the low-income communities component of this work come in?

Author Response

Response to Reviewer 3 Comments

This paper by Baek et al assesses the relationship between PM2.5 and O3 in a series of control tests of children with asthma and hospital length of stay using four years of hospital records in a S. Texas community.

The manuscript was very well prepared, the analysis was sound, the presentation was thorough, and the discussion had a good balance of implications and assessment of the data presented. It was very clear the authors put an exhaustive amount of effort into this paper submission. I do have a few minor suggestions that I think, if incorporated, can improve the manuscript.

Line 68: I did a quick google scholar search and found a number of other studies examining the relationship between air pollution and LOS. Maybe rephrase this to “only a few studies”

            => Thank you for your point. We rephrased this to “only a few studies” according to your suggestion.  

Line 124-127: Is the ozone the maximum daily 8-hr average ozone? I don’t understand the utility of an 8-hr average concentration in this analysis – was it then combined to give a daily average?

            => That is correct. Ozone data is the mean estimated 8-hour ozone concentrations in parts per billion (ppb) within 3 meters of the surface of the earth. The ozone data was attained from the Bayesian space-time downscaling fusion model called Downscaler that combines ozone monitoring data from the US Environmental Protection Agency (EPA) Air Quality System (AQS) repository of ambient air quality data and simulated ozone data from the deterministic prediction model, Models-3/Community Multiscale Air Quality (CMAQ). The CMAQ data are daily maximum 8-hour ozone concentrations calculated on a 12km*12km grid for the continental United States (Reference: Downscaler Ozone Metadata: https://data.cdc.gov/Environmental-Health-Toxicology/Daily-Census-Tract-Level-Ozone-Concentrations-2011/372p-dx3h). The CDC provides the data for PM2.5 and ozone in the census tract level.

Table 1 The Female-Low LOS cell is missing a “)”

            => Thank you for your correction. We added “)” in Table 1.

Line180: It’s important to indicate that the NAAQs have different averaging times than the data that is presented here. It’s not really an apples-to-apples comparison.

            => We agree with your point. We deleted the sentence with the NAAQS reference. We revised the numbers of references after this reference accordingly.

Table2-6: This is nitpicky, but can you include vertical lines between the pollutants? It may make the tables easier to read.

            => Thank you for your suggestion. We added vertical lines between the pollutants in Tables 2-6.

Line 210 (and throughout): Please make sure to include mention of the alpha levels on the statistical tests. The finding here, specifically, is insignificant at 0.05, but is significant at 0.10.

            => Thank you for your comment. We revised the sentences to make it clear as follows (Track-change, lines 217-221):

            “In the group of children aged 5-11 years old, the elevated PM2.5 concentration level was significantly associated with longer LOS on Lag0-1 in the single-pollutant (p=0.022) and two-pollutant (p=0.035) models. However, the ozone concentration level did not show significant associations with hospital LOS.”

Line 248-249: I don’t see how this conclusion was made from the data presented when the analysis was only for children in the first place.

            => Thanks for pointing that out. We see your point and agree. We deleted this sentence.

Line 263: I think “assess” or something to that affect Is more appropriate than “confirm”

            => We agree. We changed “confirm” to “assess”.

Line 265-266: I don’t think this statement represents the data in Table 6 completely. Sure, there aren’t any significant associations for PM2.5 but the OR is generally positive during the warm and cold season for PM2.5 as well. I just think this sentence should be rephrased.

            => Thank you for your comment. We rephrased this sentence as follows (Track-change, lines 279-281):

“Season-stratified models revealed a generally positive correlation of PM2.5 and ozone concentration levels on hospital LOS during the warm season and cold season, but significant associations were observed only for ozone.”

Line 268: comma after PM10

            => Thank you for pointing that out. We added comma (,) after PM10.

Sentences starting at 269: There is a large body of literature that finds childhood asthma hospital visits actually peaks in the Fall, despite O3 peaking in the hottest parts of summer. This has been attributed to kids being in school with other children who may transfer viruses and bacteria that can trigger an asthma attack. I think this is important context that should be included in the discussion here.

            => Thank you for your comments. As you have said, we added some information in the discussion section with additional references as follows (Track-change, lines 287-290):

            “Also, some studies presented that asthma hospital visits peak in the fall season, especially for school-aged children, despite ozone level peaking in the summer season. The explanation has been speculated to be that students transmit respiratory viruses and bacteria to their peers during the school year, causing an asthma attack [39-41].”

Line 314-317: While I agree with the statement, I don’t see this as an appropriate extension from the data presented here. Short term exposures (which would be most outdoor activities for schoolkids), proxied by the lag0 data, have some of the lowest OR correlations

            => Thank you for your point. Our study found adverse effects of outdoor air pollutants, specifically PM2.5 and ozone concentrations, on hospital LOS. Based on this finding, we are making a general suggestion that if school leaders consider the outdoor air condition when they plan/hold outdoor activities or school events, that might help reduce risk for the vulnerable students.

Line 321-322: Where did the low-income communities component of this work come in?

            => South Texas is a region that is prevalent with low-income communities. We mentioned this in the introduction with a reference (Track-change, lines 84-88):

“Therefore, the purpose of the present study is to examine the short-term effects of ambient air pollution prior to hospitalization on hospital LOS among pediatric patients with asthma in South Texas, a region with pronounced low-income communities, health disparities, and known environmental health issues [22].”

Reviewer 4 Report

Authors performed study that included retrospective data on 711 children aged 5-18 years old admitted for asthma to a children’s hospital in South Texas between 2010 and 2014. They collected air pollution data and performed the multivariate binomial logistic regression analyses to determine the association between air pollutants and hospital length of stay (LOS) on the admission day and cumulative several days before the admission, considering several confounding factors such as age, gender, ethnicity etc.

The study is of an importance, it shows significant impact of outdoor air pollution on longer LOS and indirectly on pediatric asthma control and management.

Overall, the study is well written. The results are presented clearly.

Actually I have several remarks:

  1. The drawback is the fact that only two air pollutants (PM2.5 and ozone) have been taken into analyses. However, I understand that for the retrospective study it is not sometimes possible to reach all the data. Besides authors are aware of all limitations and describe them in detailed way in the Discussion.
  2. Please add one or two sentences on the effect of ozone on respiratory system, especially asthma exacerbation to part of the discussion on the association of increased ozone concentration prior to hospital admission to prolonged hospital LOS.
  3. Can you speculate how the season or age of children could modify the relationship between outdoor air pollution and LOS?

Author Response

Response to Reviewer 4 Comments

Authors performed study that included retrospective data on 711 children aged 5-18 years old admitted for asthma to a children’s hospital in South Texas between 2010 and 2014. They collected air pollution data and performed the multivariate binomial logistic regression analyses to determine the association between air pollutants and hospital length of stay (LOS) on the admission day and cumulative several days before the admission, considering several confounding factors such as age, gender, ethnicity etc.

The study is of an importance, it shows significant impact of outdoor air pollution on longer LOS and indirectly on pediatric asthma control and management.

Overall, the study is well written. The results are presented clearly.

Actually I have several remarks:

1. The drawback is the fact that only two air pollutants (PM2.5 and ozone) have been taken into analyses. However, I understand that for the retrospective study it is not sometimes possible to reach all the data. Besides authors are aware of all limitations and describe them in detailed way in the Discussion.

            => Thank you for your comments. As you mentioned, we included only two air pollutants because of limited availability of data. We included the information in the limitation section.

2. Please add one or two sentences on the effect of ozone on respiratory system, especially asthma exacerbation to part of the discussion on the association of increased ozone concentration prior to hospital admission to prolonged hospital LOS.

            => Thank you for your suggestion. We added a sentence about the effect of ozone on respiratory system in the discussion section as follows (Track-change, lines 249-252):

“This result may be explained by the fact that exposure to ozone can adversely affect the respiratory system, including coughing, chest tightness or pain, throat irritation, and airway inflammation, especially exacerbating asthma conditions [33].”

3. Can you speculate how the season or age of children could modify the relationship between outdoor air pollution and LOS?

            => Thank you for your question. As reported in this study, we found season and age might be modifying factors for the association between outdoor air pollution and LOS. This finding is consistent with some previous research reporting. We cautiously speculate that children are more likely to be exposed to outdoor air pollution since they tend to have more outdoor activities during the warm season. It might affect the association between outdoor air pollution and LOS. However, additional studies still need to be undertaken to confirm the modifying effect of age and season.